# Genomic Instability Is Defined by Specific Tumor Microenvironment in Ovarian Cancer: A Subgroup Analysis of AGO OVAR 12 Trial

**DOI:** 10.3390/cancers14051189

**Published:** 2022-02-25

**Authors:** Jean-David Fumet, Emilie Lardenois, Isabelle Ray-Coquard, Philipp Harter, Florence Joly, Ulrich Canzler, Caroline Truntzer, Olivier Tredan, Clemens Liebrich, Alain Lortholary, Daniel Pissaloux, Alexandra Leary, Jacobus Pfisterer, Alexandre Eeckhoutte, Felix Hilpert, Michel Fabbro, Christophe Caux, Jérôme Alexandre, Aurélie Houlier, Jalid Sehouli, Emilie Sohier, Rainer Kimmig, Bertrand Dubois, Dominique Spaeth, Isabelle Treilleux, Jean-Sébastien Frenel, Uwe Herwig, Olivia Le Saux, Nathalie Bendriss-Vermare, Andreas du Bois

**Affiliations:** 1GINECO & Department of Medical Oncology, Center GF Leclerc, 1 rue du Professeur Marion, 21000 Dijon, France; 2Platform of Transfer in Cancer Biology, 21079 Dijon, France; ctruntzer@cgfl.fr; 3University of Bourgogne-Franche-Comté, 21000 Dijon, France; 4Cancer Research Center of Lyon, Université de Lyon, Université Claude Bernard Lyon 1, INSERM 1052, CNRS 5286, Centre Léon Bérard, “Cancer Immune Surveillance and Therapeutic Targeting” Team, 69000 Lyon, France; emilie.lardenois@lyon.unicancer.fr (E.L.); isabelle.ray-coquard@lyon.unicancer.fr (I.R.-C.); christophe.caux@lyon.unicancer.fr (C.C.); bertrand.dubois@lyon.unicancer.fr (B.D.); olivia.lesaux@lyon.unicancer.fr (O.L.S.); 5Leon Berard Center, Department of Pathology, 69000 Lyon, France; daniel.pissaloux@lyon.unicancer.fr (D.P.); aurelie.houlier@lyon.unicancer.fr (A.H.); isabelle.treilleux@lyon.unicancer.fr (I.T.); 6GINECO & Medical Oncology Department, Centre Léon Bérard, 28, rue Laennec, Université Claude Bernard Lyon 1, 69008 Lyon, France; olivier.tredan@lyon.unicancer.fr; 7AGO & Department of Gynecology and Gynecologic Oncology, Evang. Kliniken Essen-Mitte, 45136 Essen, Germany; p.harter@kem-med.com; 8GINECO & Department of Medical Oncology, Baclesse Cancer Center, 14118 Caen, France; f.joly@baclesse.unicancer.fr; 9AGO & Department of Gynecology and Obstetrics, Medical Faculty and University Hospital Carl Gustav Carus, Technische Universität Dresden, Dresden, Germany & National Center for Tumor Diseases (NCT), Partner Site Dresden, 01307 Dresden, Germany; ulrich.canzler@uniklinikum-dresden.de; 10Genetic and Immunology Medical Institute (GIMI), 21000 Dijon, France; 11UMR INSERM 1231, 21000 Dijon, France; 12AGO & Klinikum Wolfsburg, amO—Interdisziplinäres ambulantes Onkologiezentrum am Klieversberg, Sauerbruchstrasse 7, 38840 Wolfsburg, Germany; clemens.liebrich@klinikum.wolfsburg.de; 13GINECO & Confluent Private Hospital, Institut de Cancérologie Catherine de Sienne, 44200 Nantes, France; alain.lortholary@groupeconfluent.fr; 14Université de Lyon, Université Claude Bernard Lyon 1, INSERM 1052, CNRS 5286, Centre Léon Bérard, Cancer Research Center of Lyon, Equipe Labellisée Ligue contre le Cancer, 69000 Lyon, France; 15GINECO & Medical Oncology Department, Institut Gustave Roussy, 94805 Villejuif, France; alexandra.leary@gustaveroussy.fr; 16AGO & Zentrum für Gynäkologische Onkologie, Herzog-Friedrich-Str. 21, 24103 Kiel, Germany; jacobus.pfisterer@googlemail.com; 17INSERM U830, DNA Repair and Uveal Melanoma (D.R.U.m) PSL Research University, Institut Curie, 75005 Paris, France; alexandre.eeckhoutte@curie.fr; 18AGO & Krankenhaus Jerusalem, Moorkamp 2-6, Onkologische Tagesklinik, 20357 Hamburg, Germany; hilpert@mammazentrum.eu; 19GINECO & ICM Val d’Aurelle, oncologie médicale, 208, Avenue des Apothicaires, 34298 Montpellier, France; michel.fabbro@icm.unicancer.fr; 20Laboratory for Immunotherapy of Cancer of Lyon (LICL), Centre Léon Bérard, 69000 Lyon, France; 21GINECO & Medical Oncology Department, Hopital Cochin, 75014 Paris, France; jerome.alexandre@cch.aphp.fr; 22AGO & Charité, Medical University of Berlin, Department of Gynecology with Center of Oncological Surgery, Augustenburger Platz 1, 13353 Berlin, Germany; jalid.sehouli@charite.de; 23Synergie Lyon Cancer, Bio-Informatics Platform, 69000 Lyon, France; emilie.sohier@lyon.unicancer.fr; 24AGO & West-German Cancer Center, Department of Gynecology and Obstetrics, University of Duisburg-Essen Germany, 45136 Essen, Germany; rainer.kimmig@uk-essen.de; 25GINECO & Medical Oncology Department Centre d’Oncologie de Gentilly, 54000 Nancy, France; d.spaeth@ilcgroupe.fr; 26GINECO & Medical Oncology Department Institut de cancerologie de l’Ouest site René Gauducheau, 44800 Saint Herblain, France; jean-sebastien.frenel@ico.unicancer.fr; 27AGO & Albertinen-Krankenhaus, Department Gynecology, Süntelstraße 11a, 22457 Hamburg, Germany; uwe.herwig@immanuelalbertinen.de; 28AGO & Evangelische Kliniken Essen Mitte (KEM), 45136 Essen, Germany; prof.dubois@googlemail.com

**Keywords:** ovarian cancer, tumor immune microenvironment, HLA-E, copy number alterations, homologous recombination deficiency, HRD

## Abstract

**Simple Summary:**

Given the importance of genomic instability signatures in the management of ovarian cancer and the difficulties in defining the role of immunotherapy, our objective was to describe the tumor immune microenvironment in the light of genomic instability signatures. Intratumoral CD3^+^ T lymphocytes confirmed its prognostic value. HLA-E appears to be a robust prognostic biomarker and preferentially overexpressed in homologous recombination deficiency (HRD) ovarian cancers. Our data provide a rationale for future immunotherapy strategies targeting the inhibitory CD94/NKG2A receptor of HLA-E in HRD tumors.

**Abstract:**

Background: Following disappointing results with PD-1/PD-L1 inhibitors in ovarian cancer, it is essential to explore other immune targets. The aim of this study is to describe the tumor immune microenvironment (TME) according to genomic instability in high grade serous ovarian carcinoma (HGSOC) patients receiving primary debulking surgery followed by carboplatin-paclitaxel chemotherapy +/− nintedanib. Methods: 103 HGSOC patients’ tumor samples from phase III AGO-OVAR-12 were analyzed. A comprehensive analysis of the TME was performed by immunohistochemistry on tissue microarray. Comparative genomic hybridization was carried out to evaluate genomic instability signatures through homologous recombination deficiency (HRD) score, genomic index, and somatic copy number alterations. The relationship between genomic instability and TME was explored. Results: Patients with high intratumoral CD3^+^ T lymphocytes had longer progression-free survival (32 vs. 19.6 months, *p* = 0.009) and overall survival (OS) (median not reached). High HLA-E expression on tumor cells was associated with a longer OS (median OS not reached vs. 52.9 months, *p* = 0.002). HRD profile was associated with high HLA-E expression on tumor cells and an improved OS. In the multivariate analysis, residual tumor, intratumoral CD3, and HLA-E on tumor cells were more predictive than other parameters. Conclusions: Our results suggest HLA-E/CD94-NKG2A/2C is a potential immune target particularly in the HRD positive ovarian carcinoma subgroup.

## 1. Introduction

Epithelial ovarian cancer is the most lethal gynecologic cancer, with a 5-year survival less than 50%. Overall, more than 75% of patients are diagnosed with advanced tumor stage (FIGO stage III/IV). The current management is based on complete surgery of all macroscopic disease and platinum-based chemotherapy and maintenance therapy. Despite initial chemosensitivity, 70% of patients will relapse and finally will develop chemoresistance. Maintenance with bevacizumab and/or PARP inhibitors has shown an improvement of progression free survival (PFS) and overall survival (OS) in some subgroups [1,2,3,4,5].

High grade serous ovarian carcinoma (HGSOC) accounts for more than 75% of epithelial ovarian cancers and about 50% of HGSOC have defects in the homologous recombination DNA repair pathway, called homologous recombination deficiency (HRD). HRD can be efficiently detected as genomic instability scars using various approaches [6,7,8,9,10,11]. Recent randomized clinical trials demonstrated HRD diagnostics to be crucial for prediction of the response to PARP inhibitors impacting nowadays the first line of ovarian cancer treatment [2,5,12]. Although ovarian cancers were theoretically good candidates for immunotherapy (high expression of immunogenic tissue-specific antigens and the resulting immune infiltration is a major prognostic factor) [13,14], the systemic therapy of ovarian has improved markedly by introduction of PARP inhibitors and anti-angiogenic drugs, while the positioning of immunotherapy seems to be more difficult following the disappointing results of the main trials with these agents [15,16,17].

Immune checkpoint inhibitors (ICI) have shown modest results in main negative phase III trials [15,16,17]. These studies were negative in intention to treat (ITT) but suggested potential benefit in PFS for a subgroup analysis of PD-L1 high expression in an exploratory analysis. Beyond high PD-L1 expression (representing approximately only 20% of the whole population) it is fundamental to identify the key major actors of the anti-tumor immune response. It has been quite clearly established that tumor-infiltrating lymphocytes (TIL) in ovarian cancer are a major prognostic biomarker [18,19] but there are inconsistencies and unresolved issues in the literature concerning the prognostic and predictive significance of other immune cell infiltrates and immune pathways. A recent study reported a favorable prognostic impact of initial (before any treatment) high levels of tumor-infiltrating NK cells in HGSOC patients [20]. In a large pooled cohort and using an unbiased in silico approach, Liu et al. [21] have shown that a multitude of immune cells, such as M1 macrophages, M2 macrophages, and CD8^+^ T cells were associated with better survival of HGSOC patients treated with platinum-based chemotherapy.

Several preclinical data suggest a direct link between genomic instability and tumor immune microenvironment (TME). HRD positive score and/or BRCA-mutated status are associated with high immune infiltration in HGSOC [22,23]. Indeed, elevated levels of basal DNA damage results in the activation of the STING pathway leading to the production of type I IFN and chemoattractive chemokines (CCL5 and CXCL10) and consequently to NK cell, M1-like macrophage, and both T and B-lymphocyte recruitment in an Ag-independent manner [24]. In contrast to DNA damage, in a pan cancer analysis, Davoli et al. have suggested that somatic copy number alteration, called aneuploidy, was correlated with immune evasion markers and inversely correlated with patient’s response to immunotherapy [25]. Furthermore, combined analysis of genomic instability and immune parameters could have a prognostic value as shown by Morse et al. who have recently published that patients with both CD3^+^ T lymphocytes^high^ and HRD profile presented the best prognostic value [26].

Using a subset of patients from AGO OVAR 12 trial [27], we sought to characterize the prognostic and predictive impact of genomic instability scores in HGSOC patients and relationships with immune response parameters.

## 2. Materials and Methods

### 2.1. IHC Analysis

This cohort of HGSOC patients constitutes an excellent basis for exploring several concepts with non-pretreated patients who have benefited from a standard chemotherapy plus or minus nintenanib supervised by the clinical trial. Samples are of high quality with a cellularity suitable for the analysis of biomarkers. A tissue microarray (TMA) was performed on all patients’ samples (3 TMA per slide per patient); specimens were analyzed at the Leon Bérard research center. TMA construction based on tissue cylinders with a diameter of 0.6 mm each were taken from representative tumor areas of selected tumor tissue blocks by a pathologist (IT). To evaluate the location of CD3^+^ cells (tumor vs. stroma), the CD3 staining was analyzed in the whole slides. Primary antibody specific for CD3 (clone VENTANA-ROCHE, 790-4341 E01439), CD8 (clone VENTANA-ROCHE, 790-4460 E08164), CD20 (DAKO, M0755 00094151), CD163 (LEICA-NOVOCASTRA 32265 6027910), IgA (DAKO, A0262 00089632), IgG (DAKO, A0423 20003793), ICOS (SPRING BIOSCIENCES, M3984 41030LVA), CD73 (Cell Signaling #13160 1; 06/15), PD-L1 (DAKO, 28.8), FOXP3 (ABCAM, ab20034-GR170762), CD39 (ABCAM, 22A9 ab49580), MXA (ABNOVA H00004599-B01P), DC-LAMP (DENDRITICS, DDX0191P-DDX0191P-027), BDCA2 (DENDRITICS DDX0041-DDX0041-022), HLA-E (EXBIO, MEM-E/02), CDK12 (IgG SIGMA HPA08038), and NKp46 (INNATE PHARMA MOS2-M-H46-8E5B-IC4) was applied.

CD3, CD163, PDL1, HLA-E, NKp46, and CDK12 stainings were estimated by a semi quantitative assessment: 0 = no staining, 1 = few number of cells stained, 2 = moderate number of cells stained, 3 = high number of cells stained. Absolute quantification was achieved for CD8, CD20, IgA, IgG, ICOS, and FOXP3.

Binary stratification between positive and negative was performed for CD39 lymphocytes, CD39 vessels (only in TMA with vessels, other TMA were excluded for analysis), DC-Lamp, and BDCA2.

For MXA, we performed a H score based on the addition of (percentage of tumor cells stained with low density × 1) + (percentage of tumor cells stained with moderate density × 2) + (percentage of tumor cells stained with high density × 3) + (percentage of tumor cells non stained × 0).

Each cutoff for high or low expression was determined for each biomarker using a best cutoff strategy. Thus, depending on the survival criteria, the cutoff can vary. All the cutoffs used are available in the Appendix A.

### 2.2. CGH Analysis

DNA extraction was performed by macro-dissecting formalin-fixed paraffin-embedded tissue block sections followed by the use of the QIAamp DNA micro kit (Qiagen #56304, Hilden, Germany). Fragmentation and labeling were done according to manufacturer’s protocol (Agilent Technologies, Santa Clara, CA, USA), using 1.5 μg of genomic DNA. Tumor DNA was labeled with Cy5, and a reference DNA (Promega #G1521or #G1471, Madison, WI, USA) was labeled with Cy3. Labeled samples were then purified using KREApure columns (Agilent Technologies #5190-0418). Labeling efficiency was calculated using a Nanodrop ND2000 Spectrophotometer. Co-hybridization was performed on 4 × 180 K Agilent Sureprint G3 Human whole-genome oligonucleotide arrays (Agilent Technologies #G4449A). Slides were washed, dried, and scanned on the Agilent SureScan microarray scanner. Scanned images were processed using Agilent Feature Extraction software V11.5 and the analysis was carried out using the Agilent Genomic Workbench software V7.0 and a custom analysis pipeline allowing baseline corrections and generating biological status for the detected segments.

The custom analysis pipeline is based on R packages limma [28] and ArrayTV [29] for the normalization step, the R package DNAcopy for the segmentation step, and an in-house pipeline to define segments status (normal, gain, loss…) depending on logratio threshold and to generate pangenome and chromosome plots. Associated with a web interface, R Shiny [30] allowed biologists to visualize the plot with the status for each segment, validate the results, and eventually recalibrate it by corrected position of the baseline and generate the new output files with corrected segments status.

Genomic index was calculated as follows: GI = A^2^/C where A corresponded to the total number of alterations (segmental gains or losses) and C to the number of chromosomes affected by these alterations.

Somatic copy number alterations (SCNA) scores were computed by adapting Davoli et al. methodology [25] to the specificity of CGH array using amplification/deletion/normal status from segmentation files; scores were computed at the focal, arm, and chromosome levels. Focal level concerns deletions or amplifications involving a region smaller than 50% of a chromosome arm, chromosome level concerns all cases where both arms of a chromosome had the same copy number change (in value and sign), and arm level concerns all other situations. Scores correspond to the total level of deletions or amplifications at each of these levels.

The determination of the HRD status was performed by shallowHRD [31] with minor adaptation to CGH profiles. shallowHRD is an adaptation of the Large-scale State Transition [6] approach to the CNA profiles lacking allelic counts information. CGH profiles segmented by DNAcopy were formatted to shallowHRD input format and further processed providing Large Genomic Alterations (LGA) counts for each profile. Briefly shallowHRD pipeline consists in (i) detecting a cut-off representing a one copy difference for each profile; (ii) smoothing the profile in a step-wise manner to obtain a robust and non-redundant segmentation, proceeding mainly by merging adjacent large segments with the shift less than the cut-off; (iii) integrating the small segments following the same strategy. LGAs representing copy number breaks between the large segments (>10 Mb) are called on the final segmentation after filtering small interstitial CNAs. HRD is estimated based on the number of LGAs called as described in [31] with the threshold for HRD set to 18.

It should be noted that the germline or somatic BRCA mutation status was not explored and reported at the time of this clinical trial.

### 2.3. Statistical Analysis

Patient characteristics were compared using the chi-square test for qualitative variables and the Wilcoxon test for continuous variables, as appropriate.

Cox regression models were used to estimate hazard ratios (HR) and 95% confidence intervals (CIs) for progression-free survival (PFS) and overall survival (OS). Survival curves were estimated by the Kaplan–Meier method and compared with the Log-rank test (univariate analysis). Optimal cutoffs for continuous variables scores were chosen based on a maximally selected rank statistics [27].

Statistical tests were two-sided, and a *p*-value < 0.05 was considered statistically significant. Data analysis was performed using R statistical software and presented with Prism 7.02 (GraphPad, San Diego, CA, USA).

Multivariate Cox models were generated using Cox regression with lasso penalty for progression-free (PFS) or overall (OS) survival. This model allows both variables selection and HR estimation in cases of several potential markers to test in the model relatively to the number of observations. To limit optimism and overfitting, 500 bootstrap samples were generated and only variables mostly (>2/3) selected through the 500 corresponding lasso Cox models were kept in the final models. This cut-off was chosen empirically. A multivariate Cox model with lasso penalty was estimated involving clinical, immunological, and genomic variables with *p*-values < 0.1, as estimated through univariate Cox models.

## 3. Results

### 3.1. Patient Population

Tumor samples from German and French patients included in the phase III trial AGO OVAR 12 were retrospectively retrieved [27]. This phase 3 trial included chemotherapy-naive patients with International Federation of Gynecology and Obstetrics (FIGO) IIB-IV ovarian cancer and upfront debulking surgery that were randomly assigned (2:1) to receive six cycles of carboplatin and paclitaxel followed by maintenance nintedanib (a tyrosine kinase inhibitor targeting VEGFR) or placebo. The current cohort selecting available tumor samples for 125 patients included 103 high grade serous ovarian carcinoma (HGSOC) available for translational research from GINECO group and AGO/German centers. The rate of upfront surgery with no residual disease in this subgroup was 52.4% similar to the whole population. The most relevant patient’s characteristics are presented in Table 1. No difference in PFS and OS was observed according to treatment group (nintedanib or placebo) in this subgroup compared to the whole population (Appendix A).

### 3.2. Intratumoral CD3 Confirmed to Be a Major Prognostic Biomarker in HGSOC

A large panel of 21 biomarkers was assessed by immunohistochemistry (IHC). Intratumoral CD3^+^ lymphocytes infiltration was the only statistically significant prognostic biomarker associated with a longer PFS (Figure 1A) with a median of 32 months (95% CI [28.1; 53.1]) in CD3^high^ versus 19.6 months (95% CI [16.6; 28.4]) in CD3^low^ (HR = 0.52 [0.32; 0.85], *p* = 0.009) (Figure 1B) and a longer OS (HR = 0.27 [0.11; 0.65], *p* = 0.003) (Figure 1C,D). As previously published [13], stromal CD3^+^ lymphocytes infiltration was not prognostic for PFS (HR = 1 [0.63; 1.59], *p* = 0.51) and OS (HR = 0.69 [0.36; 1.32], *p* = 0.7). At the opposite, CD73^positive^ vessels tended to be associated with a worse PFS (median PFS of 21.4 months vs. 32 months, HR = 1.64 [0.95; 2.82], *p* = 0.07) (Figure 1A).

### 3.3. HLA-E on Tumor Cells Is an Emergent Prognostic Biomarker in HGSOC

Among 98 assessable samples, 73.5% (*n* = 72) overexpressed HLA-E (scoring 2 and 3, called HLA-E^high^) and 26.5% (*n* = 26) expressed normally or low HLA-E (scoring 0 and 1, called HLA-E^low^) on tumor cells. Figure 2A reports examples of histological staining for HLA-E expression on TMAs.

HLA-E overexpression was not correlated with any clinical prognostic factors such as FIGO (Chi2 test, *p* = 0.36), performance status (Chi2 test, *p* = 0.92), age (Wilcoxon test, *p* = 0.68), and residual tumor (Chi2 test, *p* = 0.75). However, HLA-E was significantly associated with a longer OS (median OS not reached vs. 52.9 months, HR = 0.36 [0.18; 0.72], *p* = 0.002) (Figure 1C and Figure 2B) and a trend was observed for HLA-E as prognostic biomarker of improved PFS (HR = 0.70 [0.42; 1.18]) (Figure 1A). Furthermore, platinum-sensitive tumors (tumors from patients relapsing more than 6 months after the last platinum-based chemotherapy cycle) were enriched in HLA-E expression. Indeed, 75.5% (*n* = 67) of platinum sensitive relapsed patients exhibited HLA-E^high^ tumor cells whereas 55.5% (*n* = 5) of platinum-resistant patients exhibited HLA-E^high^ tumor cells (Figure 2C). We also observed that HLA-E^high^ patients were enriched in intratumoral CD3^+^ infiltration (43% (*n* = 31) of intratumoral CD3^high^ in HLA-E^high^ patients versus 23.1% (*n* = 6) of intratumoral CD3^high^ in HLA-E^low^ patients, Fisher test *p* = 0.09), FOXP3 (44.4% (*n* = 32) of FOXP3^high^ in HLA-E^high^ patients versus 15.3% (*n* = 4) of FOXP3^high^ in HLA-E^low^ patients, Fisher test *p* = 0.009), IgG (43% (*n* = 31) of IgG^high^ in HLA-E^high^ patients versus 15.4% (*n* = 4) of IgG^high^ in HLA-E^low^ patients, Fisher test *p* = 0.02), and ICOS (38.9% (*n* = 28) of ICOS^+^ in HLA-E^high^ patients versus 7.7% (*n* = 2) of ICOS^+^ in HLA-E^low^ patients, Fisher test *p* = 0.003), compared to HLA-E^low^ patients (Figure 2D). Survival analysis for intratumoral CD3 infiltration separately in HLA-E^low^ and HLA-E^high^ patients was performed. The positive prognostic impact on PFS and OS of intratumoral CD3^+^ lymphocytes remains strongly significant in the HLA-E^low^ subgroup (HR = 0.27 [0.11; 0.67], *p* = 0.004 and HR = 0.16 [0.05; 0.5], *p* = 0.04 respectively). Whereas, in the good prognostic HLA-E^high^ subgroup, the addition of intratumoral CD3^+^ lymphocytes infiltration did not improve the prognostic impact of HLA-E (Figure 2E). Furthermore, we noted that HLA-E^high^ subgroup was not enriched in NKp46^+^ cells and that there is no added prognostic value when combining HLA-E and NKp46 compared to HLA-E alone (Appendix A).

### 3.4. Genomic Instability Confirmed to Be Correlated to Survival in HGSOC

Genomic instability was analyzed by CGH using different published scores: evaluation of focal Somatic Copy Number Alteration (SCNA), chromosome arm and whole chromosome SCNA [25], Genomic Index (GI) [32], and HRD score [6,31] and we assessed its prognostic impact. The distribution of patients according to these different scores is presented in Figure 3A. Among genomic instability parameters, only the high GI index defined by an optimal cutoff of 88 was statistically associated with poor PFS (HR = 1.65 [1.04; 2.63], *p* = 0.03) (Figure 3B). Focal SCNA tended to be associated with poor PFS (HR = 1.5 [0.93; 2.39], *p* = 0.09) (Figure 3B). There was no impact of SCNA regarding chromosomal arm and whole chromosome levels on PFS (Figure 3B). Again, focal SCNA tended to be associated with poor OS (HR = 2.15 [0.9; 5.17], *p* = 0.08) (Figure 3C). HRD profile did not impact PFS but was significantly associated with a good OS (HR = 0.36 [0.15; 0.84], *p* = 0.02) (Figure 3C). Other signatures (GI and chromosome arm SCNA) were not identified to correlate to OS (Figure 3C).

### 3.5. Relationship between Genomic Instability and Tumor Immune Microenvironment for HGSOC

The links between the previous genomic instability scores and the TME were explored. HRD tumors were significantly enriched in HLA-E expression compared with non-HRD (called HRP for homologous recombination proficient) tumors (89% vs. 44%, Chi-squared test *p* = 0.005) (Figure 4A). Furthermore, IFN-related signature MXA was significantly higher in HRD tumors compared to HRP tumors (Wilcoxon test *p* = 0.017) (Figure 4B). Furthermore, GI^low^ tumors were significantly associated with the presence of CD39^positive^ vessels (75.7% in GI^low^ vs. 54% in GI^high^, Chi-squared test *p* = 0.05) (Figure 4C). Finally, high focal SCNA was significantly associated with higher rate of intratumoral CD3 (0.41 vs. −0.24, *p* = 0.06) and CD20 (0.38 vs. −0.48, *p* = 0.006) (Figure 4D,E) while a high arm SCNA was associated with higher rate of CD20 (0.22 vs. 0.03, *p* = 0.04), CD163 (0.24 vs. −0.44, *p* = 0.03), and FOXP3 (0.70 vs. 0.03, *p* = 0.03) (Figure 4F–H). All results are available in Appendix A.

### 3.6. Multivariate Analysis

For PFS analysis, following variables were selected by the lasso model: residual tumor (HR = 1.44; 95% CI: [1.00; 2.46]), intratumoral CD3 (HR = 0.66; 95% CI: [0.39; 1.00]), CD73 vessels (HR = 1.50; 95% CI: [1.00; 2.78]), and NKp46 (HR = 0.65; 95% CI: [0.29; 1.00]). Concerning OS, residual tumor (HR = 2.19; 95% CI: [1.00; 13.30]) and HLA-E (HR = 0.23; 95% CI: [0.22; 1.00]) were the only two independent factors identified by the lasso model (Table 2).

## 4. Discussion

Using a cohort of 103 HGSOC patients extracted from a large randomized phase III trial [27], we performed an extensive analysis of their TME by IHC showing intratumoral CD3^+^ T lymphocytes as a strong positive prognostic biomarker as widely published [19]. Furthermore, HLA-E on tumor cells was identified as an independent positive prognostic biomarker in HGSOC. HLA-E is a non-classical MHC class I molecule expressed by different cells including tumor cells. It serves as a ligand to CD94/NKG2A, a major immune checkpoint receptor expressed on subsets of NK cells and some activated CD8^+^ T lymphocytes [33]. The binding of HLA-E to CD94/NKG2A receptor on NK cells and T cells transduces inhibitory signals that suppress effector functions, such as cytotoxicity [34], whereas the binding of HLA-E to CD94/NKG2C transduces stimulatory signals through recruitment of the intracellular adaptor DAP-12 [35]. Although classical HLA alleles are frequently lost in human cancer to prevent T-cell recognition [36], the upregulation of HLA-E is a common feature in several cancer entities including ovarian cancer [37]. In line with HLA-E overexpression described in microsatellite instable tumors in colorectal cancer [38], we also reported a higher prevalence of HLA-E overexpression in HRD tumors that are known to be more immune infiltrated, due to their genomic instability. These data suggest that beyond genomic instability, genotoxic stress (by MMR deficiency or HR deficiency) could induce hyper-expression of HLA-E. Future studies will be necessary to determine if the overexpression of HLA-E by tumor cells is secondary to IFN-γ secretion in these highly infiltrated tumors [39,40] or secondary to DNA damage that may lead to type I IFN production [41]. Indeed, HLA-E on freshly isolated ovarian cancer cells was up-regulated by IFN-γ treatment [39], while the role of type I IFN in the regulation of HLA-E expression is still an open question. In agreement with previous data [42], we observed that HLA-E-overexpressing tumors were highly enriched in Treg (FOXP3+, ICOS+) and IgG. Moreover, there is no added benefit of T-cell infiltration in HLA-E overexpressed tumor, in agreement with a previous work by Gooden et al. [43]. We therefore hypothesize that targeting HLA-E pathways in these “hot” tumors could trigger T lymphocytes and NK cells and sensitize them to ICI, as suggested in preclinical models [44]. Targeting CD94/NKG2A rather than HLA-E molecule with antagonist molecules could be key to unlock the CD94/NKG2A-mediated inhibition and to preserve the CD94/NKG2C positive signaling in cytolytic lymphocytes (CD8 T cells, NK cells, and NKT cells) infiltrating ovarian cancers. From a therapeutic perspective, humanized monoclonal antibodies targeting NKG2A (Monalizumab) are readily available and currently in clinical trials for cancer, hence representing a promising strategy for future clinical studies in HLA-E positive ovarian tumors.

Beyond HRD, we showed that CGH-based genomic analysis has defined the GI as a potential novel negative prognostic biomarker in HGSOC. Indeed, high GI was associated with a poor PFS, without impacting OS. These results are in accordance with previous data of poor clinical outcome in high chromosomal instability tumors [32]. Using the A^2^/C formula to calculate the GI, a high GI corresponds to a high number of genomic alterations on a limited number of chromosomes. In contrast to the HRD profile which corresponds to extensive scarring on all chromosomes, the GI represents a form of focal aneuploidy which seems to be associated with a poor prognosis. These genomic instability scores should be tested in larger cohorts to validate their prognostic value as well as their predictive value for response to PARP inhibitors.

We acknowledge certain limitations of this subgroup’s analysis including only around one hundred of patients. Not all samples were available for CGH analysis and final results concern only a part of the cohort. The calculation of the HRD score was performed on CGH data not considering ploidy or tumor purity. This method gave an estimate of genomic instability. Finally, BRCA status was not available at the time of the study time. These limitations notwithstanding, our findings seem clinically and translationally relevant and will have to be confirmed in future larger series.

## 5. Conclusions

Our study confirmed differential TME focusing on HLA-E expression on tumor cells between HRD and HRP HGSOC, which argues in favor of a need for a stratified strategy in future clinical trials. We identified a strong link between HRD and HLA-E expression that brings a promising strategy through CD94/NKG2A targeting in HRD positive ovarian cancer.

## Figures and Tables

**Figure 1 cancers-14-01189-f001:**
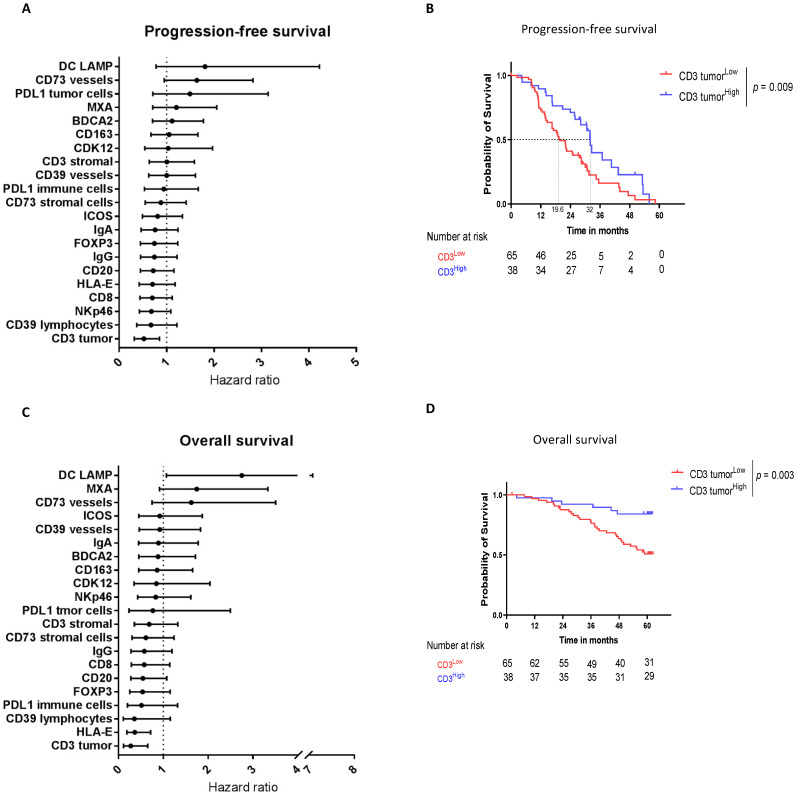
Intratumoral CD3 confirmed to be a major prognostic biomarker in HGSOC. Intratumoral CD3 (“CD3 tumor”) is the main prognostic biomarker of HGSOC patients’ survival. (**A**–**C**) Forest Plots of the univariate analysis showing the hazard ratio for (**A**) progression-free survival and (**C**) overall survival for each immune parameter evaluated by IHC. (**B**–**D**) Kaplan–Meier estimates for (**B**) progression free survival and (**D**) overall survival according to the intratumoral CD3 expression using the best cutoff.

**Figure 2 cancers-14-01189-f002:**
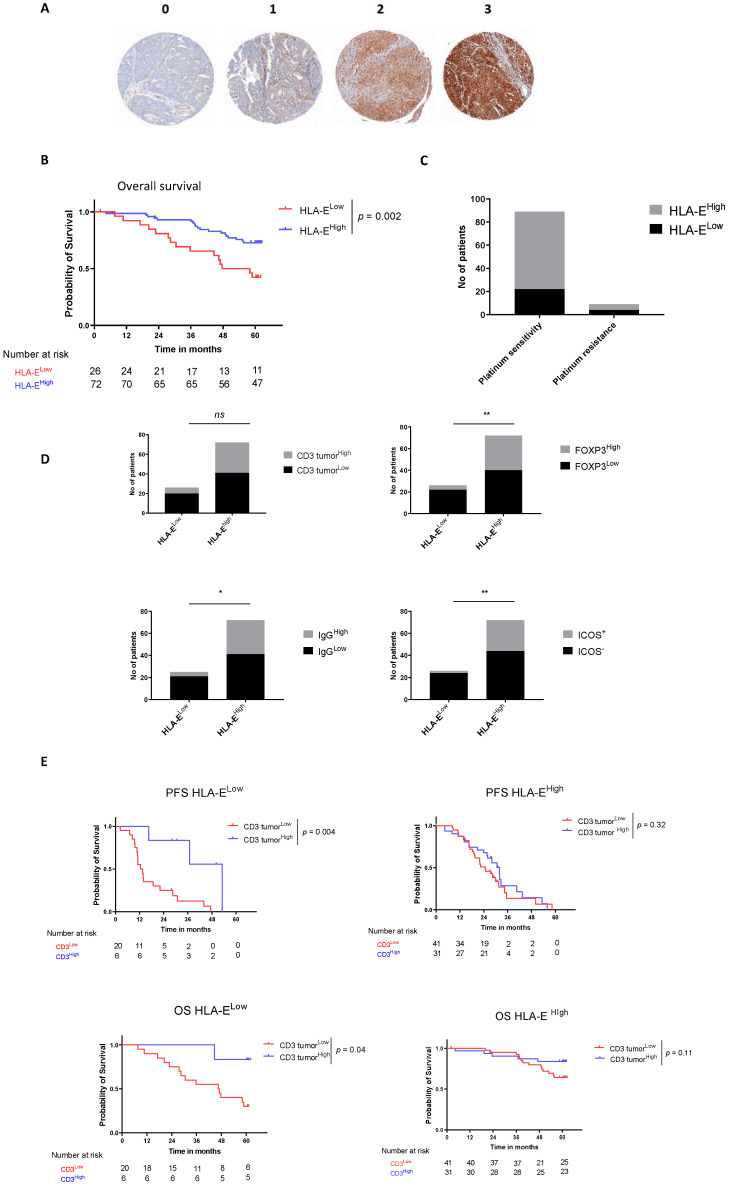
HLA-E on tumor cells is an emergent prognostic biomarker in HGSOC. HLA-E predicts HGSOC patients’ survival. (**A**) Representative snapshots of HLA-E staining assessed by IHC in percentage of positive tumor cells where 0 is lower than 1%, 1 is between 1% and 5%, 2 between 5% and 50%, and 3 is higher than 50%. 0 and 1 scoring corresponded to HLA-E^low^ expression. 2–3 corresponded to HLA-E^high^ expression. (Original magnification ×10). (**B**) Kaplan–Meier estimates for overall survival according to the HLA-E expression using the best cutoff. (**C**) Bar plots showing the proportion of HLA-E expression stratified according to the platinum sensitivity (platinum sensitive vs. platinum resistant). (**D**) Bar plots showing the repartition of immune populations (intratumoral CD3, Foxp3, IgG, and ICOS) according to the expression of HLA-E. (**E**) Kaplan–Meier curves for PFS (upper panels) and OS (lower panels) according to the HLA-E and intratumoral CD3 expression. * *p* < 0.05, ** *p* < 0.005.

**Figure 3 cancers-14-01189-f003:**
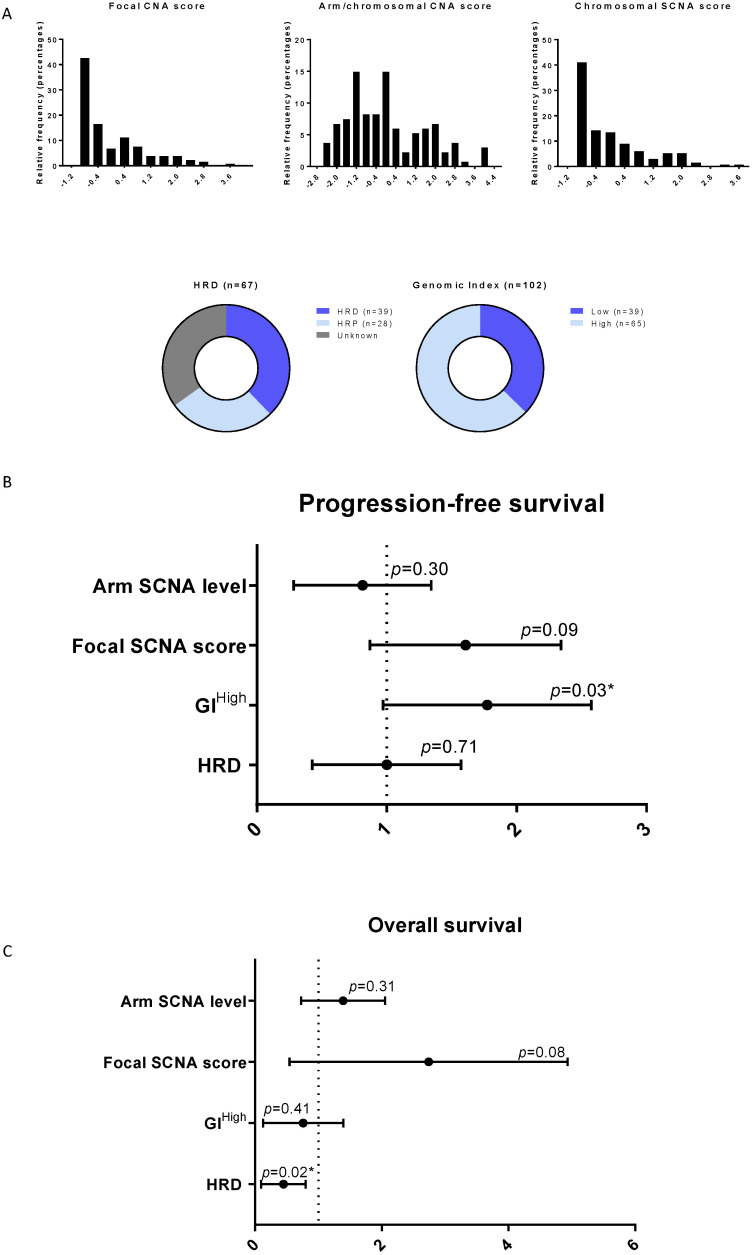
Prognostic and predictive value of genomic instability signatures. (**A**) Repartition of patients according to genomic scores: histograms for Focal and Arm/chromosomal SCNA scores. Donut charts showing percentages of HRD (*n* = 39), HRP (*n* = 28) or unknown (*n* = 36) patients and percentages of GI^high^ or GI^low^ patients that were stratified by best cutoff of 88. (**B**) Forest Plots showing the hazard ratio for progression free survival for each genomic signature. (**C**) Forest Plots showing the hazard ratio for overall survival for each genomic signature. * *p* < 0.05.

**Figure 4 cancers-14-01189-f004:**
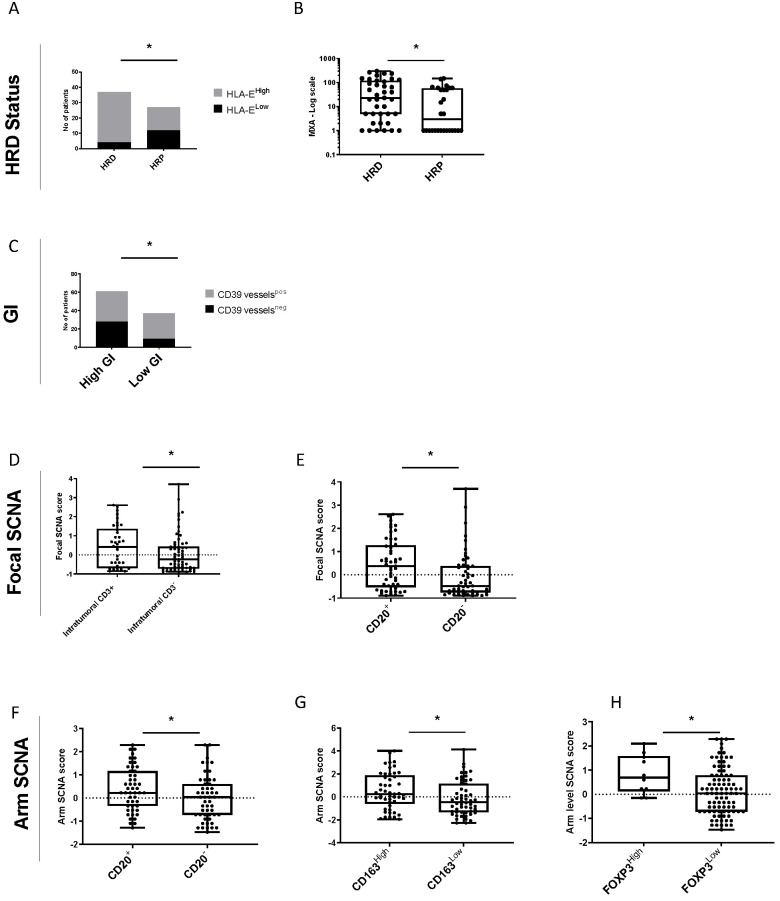
Genomic instability and tumor immune microenvironment in HGSOC. (**A**) Bar plots showing the proportions of HLA-E^high^ versus HLA-E^low^ patients according to HRD status (HRD versus HRP). (**B**) Boxplots showing the MXA score according to HRD status (HRD versus HRP). (**C**) Bar plots showing the proportions of patients with CD39 vessels^pos^ versus CD39 vessels^neg^ according to GI status (GI^high^ versus GI^low^). (**D**,**E**) Boxplots showing the Focal SCNA score according to (**D**) intratumoral CD3 expression and (**E**) CD20 expression. (**F**–**H**) Boxplots showing the Arm SCNA score according to (**F**) CD20 expression, (**G**) CD163 expression, and (**H**) FOXP3 expression. * *p* < 0.05.

**Table 1 cancers-14-01189-t001:** Baseline demographics.

Characteristic	All (*n* = 103)
median age	58.8 (44; 72.9)
FIGO	
IIB	6 (5.8%)
IIC	4 (3.9%)
IIIB	6 (5.8%)
IIIC	57 (55.4%)
IV	30 (29.1%)
Histology	
High grade serous	103 (100%)
Optimal cytoreduction	
yes	49 (52.4%)
no	54 (47.6%)
Performance status	
0	64 (62.1%)
1	36 (35%)
2	3 (2.9%)
Treatment	
nintedanib	70 (68%)
placebo	33 (32%)

**Table 2 cancers-14-01189-t002:** Univariate and multivariate analysis.

	Progression-Free Survival			Overall Survival		
	Univariate Analysis		Multivariate Analysis	Univariate Analysis		Multivariate Analysis
	HR (95% CI)	*p*-Value	HR (95% CI)	HR (95% CI)	*p*-Value	HR (95% CI)
**Clinical**						
FIGO (IV vs. others)	1.57 [0.96; 2.55]	0.07		2.277 [1.19; 4.37]	0.02	
Age (>60 vs. ≤60 y)	1.15 [0.72; 1.83]	0.55		1.41 [0.74; 2.70]	0.3	
Complete cytoreduction CC-0 (Yes or no)	0.45 [0.29; 0.73]	<0.001	0.69 [0.41; 1.00]	0.31 [0.15; 0.64]	<0.001	0.46 [0.08; 1.00]
Performance status (0 vs. 1–2)	1.48 [0.93; 2.35]	0.1		1.38 [0.72; 2.64]	0.3	
Treatment (Placebo vs. Nintedanib)	0.6 [0.35; 1.03]	0.07		0.9 [0.44; 1.82]	0.78	
**Immunological** *						
BDCA2	1.12 [0.7; 1.78]	0.65		0.88 [0.45; 1.72]	0.72	
CD163	1.05 [0.67; 1.66]	0.82		0.86 [0.45; 1.66]	0.66	
CD20	0.72 [0.45; 1.15]	0.17		0.55 [0.28; 1.08]	0.08	
CD3 stromal	1 [0.63; 1.59]	0.51		0.69 [0.36; 1.32]	0.70	
CD3 tumor	0.52 [0.32; 0.85]	0.01	0.66 [0.39; 1.00]	0.27 [0.11; 0.65]	0.004	
CD39 lymphocytes	0.67 [0.37; 1.22]	0.19		0.92 [0.46; 1.83]	0.09	
CD39 vessels	1 [0.62; 1.61]	1.00		0.36 [0.11; 1.16]	0.82	
CD73 stromal cells	0.88 [0.55; 1.41]	0.79		0.61 [0.3; 1.24]	0.34	
CD73 vessels	1.64 [0.95; 2.82]	0.08	1.50 [1.00; 2.78]	1.62 [0.75; 3.51]	0.22	
CD8	0.7 [0.44; 1.12]	0.13		0.58 [0.29; 1.15]	0.12	
CDK12	1.03 [0.54; 1.97]	0.93		0.85 [0.35; 2.04]	0.71	
DC LAMP	1.81 [0.78; 4.22]	0.17		2.75 [1.07; 7.08]	0.04	
FOXP3	0.74 [0.45; 1.24]	0.25		0.54 [0.25; 1.15]	0.11	
HLA-E	0.7 [0.42; 1.18]	0.18		0.36 [0.18; 0.72]	0.004	0.23 [0.02; 1.00]
ICOS	0.81 [0.49; 1.34]	0.41		0.92 [0.46; 1.87]	0.82	
IgA	0.76 [0.46; 1.24]	0.27		0.89 [0.45; 1.78]	0.75	
IgG	0.74 [0.45; 1.22]	0.24		0.58 [0.28; 1.19]	0.14	
MXA	1.21 [0.71; 2.06]	0.49		1.75 [0.92; 3.33]	0.09	
NKp46	0.68 [0.43; 1.09]	0.1	0.65 [0.29; 1.00]	0.83 [0.43; 1.62]	0.59	
PD-L1 (immune cells)	0.94 [0.53; 1.67]	0.84		0.51 [0.2; 1.32]	0.17	
PD-L1 (tumor cells)	1.49 [0.71; 3.14]	0.29		0.77 [0.24; 2.5]	0.66	
**Genomic**						
HRD status (HRD vs. HRP)	0.89 [0.49; 1.62]	0.7		0.36 [0.15; 0.84]	0.02	
GI (Low vs. High)	0.59 [0.36; 0.97]	0.03		0.41 [0.35; 1.5]	0.41	
Focal CNA score	1.5 [0.93; 2.39]	0.09		2.15 [0.9; 5.47]	0.08	
Arm CNA level	0.7 [0.35; 1.39]	0.3		1.28 [0.79; 2.10]	0.31	

* For binary variables, coefficients are presented for High vs. Low expression.

## Data Availability

All results are contained within the article. All data are available on request from the corresponding author.

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
