# Peer review of "Genomic Instability Is Defined by Specific Tumor Microenvironment in Ovarian Cancer: A Subgroup Analysis of AGO OVAR 12 Trial"

_cancers, 2022, doi:10.3390/cancers14051189_

Round 1

Reviewer 1 Report

This study examines the prognostic value of genomic instability assessment via SNP-aCGH relative to tumor immune microenvironment as assayed by IHC in tissue microarrays from a subset of 103 HGSOC patients of the AGO OVAR 12 phase III trial. The selected patients received carboplatin and paclitaxel chemotherapy combined to maintenance via a VEGFR inhibitor or placebo. The authors focus on the association between high HLA-E expression in HGSOC tumor samples and the efficiency of homologous recombination mediated DNA repair via the examination of a series of surrogate markers of genomic instability. The study proposes the HLA-E/CD94-NKG2A/2C axis as a potential immune target in the HR-Deficient ovarian carcinoma subgroup. There are several issues concerning this study: the number of patients is limited, the number of patients examined by aCGH is unclear, the status of BRCA1-2 mutations is unknown and the biological relationship between the high expression of HLA-E and homologous recombination mediated DNA repair in this histopathological type of tumors remains obscure. Although the manuscript cannot be accepted in the present form for publication in Cancers it may improve by extensive revisions.

Major concerns:

  • The authors must define the BRCA mutation status at least based on the aCGH analysis (deletions of BRCA1 or BRCA2 genes).
  • Although it is mentioned in the text (line:311) the ploidy and heteroploidy indices do not appear anywhere in the results or discussion.
  • The authors must discuss the discrepancy between HLA-E overexpression in microsatellite repair deficient colorectal cancers that are characterized by extremely low chromosomal instability and the higher prevalence of HLA-E overexpression in HRD HGSOC that are expected to present increased chromosomal instability.
  • A more comprehensive presentation and discussion of the genomic instability data per parameter and number of patients per parameter is necessary.
  • A patient stratification based on the type and chromosome location of shared imbalances, HR-efficiency and TME will be highly informative.
  • The text has a high number of acronyms and abbreviations an explanatory table would be beneficial to the reader.

Minor Comments:

Line 103: “…the systemic therapy of ovarian has improved…” the phrase is incomplete

Lines 194-195 “…Genomic index was calculated as follows: GI = A2/C where A corresponded to the total number of alterations (segmental gains or losses) and C to the number of chromosomes affected by these alterations…” The formula must be correctly written.

Reviewer 2 Report

This is a novel, very interesting and comprehensive study on a cohort of 103 patients with high grade serous ovarian carcinoma (HGSOC) from a large randomized phase III trial. The authors sought to determine the key major contributors of the antitumor immune response in HGSOC in an effort to find new strategies beyond using inhibitors of the PD-1/PD-L1 pathway which have been disappointing in this type of cancer. The study confirmed intratumoral CD3 to be a major prognostic biomarker in HGSOC and identified HLA-E on tumor cells as an emergent prognostic biomarker in HGSOC. In addition, genomic instability, particularly the homologous recombination deficiency (HRD) profile was associated with high HLA-E expression on tumor cells and an improved overall survival. These new findings are very significant suggesting that potential blockade particularly of the inhibitory receptor CD94/NKG2A on NK cells may benefit antitumor responses in HGSOC, thus the study opens new opportunities for promising therapeutic interventions. The authors discuss the significance of these results and acknowledge certain limitations of the study.

I only have few minor suggestions for the text:

Line 62: Add (+) symbol: …tumoral CD3+ T lymphocytes…

Line 65: Separate the words: …of HLA-E in HRD tumors…

Line 103: Correct to: …of ovarian cancers has improved…

Line 141: Maybe change to: …plus or minus…

Line 192: Correct to: …by corrected position…

Line 213: There is a gap at end of phrase.

Line 249: …to the whole population…

Line 258: The reference is missing: ...as previously published…

Line 389: …humanized…

Line 402: …PARP inhibitors.

Line 405: Correct phrase, maybe write: …BRCA status was not available at the time of the study” (I think this is what you meant)

Round 2

Reviewer 1 Report

Despite the answers provided by the authors in reviewer’s comments and some minor changes in the revised version of this manuscript, there are still important concerns regarding the design of the study, the clarity of the methodology used and the quality of the presented results rendering the manuscript not suitable for publication in Cancers in its present form.

Major comments

  • The number of tumors examined by aCGH still remains unclear, are these 67 (as indicated by the HRD/HRP results in revision cover letter) or 103?
  • The authors declare that they wanted to analyze the ploidy status of the tumor samples, but they claim that these data were not found relevant and therefore they are not included in original or revised version. Nevertheless, chromosome number could be estimated by the sum of the copy numbers detected at the pericentric regions. In addition, according to the revised text (lines 198-200) “…Somatic copy number alterations (SCNA) scores were computed by adapting Davoli et al. methodology [20] using segmentation files resulting from CGH analysis; scores were computed at the focal, arm and chromosome levels…”. However, in Davoli et al. 2017, it is clearly stated that in order to determine SCNA calls, i.e., the presence or absence of segmental, arm or whole chromosome amplifications or deletions, they considered different noise thresholds based on tumor purity and ploidy index following the formula: R = (p∗q + 2∗(1 − p))/(p∗T + 2∗(1 − p))(where ploidy=T, the relative copy number of a certain region=R, in and the integer copy number of that region=q). The authors need to clarify if they used this formula and if they arbitrarily considered diploidy, triploidy or other ploidy index to calculate SCNA calls, and if and how they estimated the purity of the samples.
  • There are more unclear statements about the methodology used to assay aneuploidy: In lines 98-100, it is stated:”… HRD profile defined in the recent randomized clinical trials as genomic instability scars through loss of heterozygosity (LOH), telomeric allelic imbalance (TAI), and large-scale state transitions (LST)…”. It is noteworthy that no results on TAI, or LOH are presented while ploidy index or median chromosome number are necessary to calculate LST. In lines 312-314 the authors state: “…Genomic instability was analyzed by CGH using different published scores: evaluation of focal Somatic Copy Number Alteration (SCNA), chromosome arm and whole chromosome SCNA [20], Genomic Index (GI) [28], and HRD score [29] ...”. However, in reference #28 (Bertucci et al 2018) the Genomic Grade Index (GGI) is defined as a 108-gene expression signature while in Morse et al HRD was defined as a damaging mutation in one of 12 genes in the homology mediated DNA repair pathways or promoter hypermethylation in BRCA1 or RAD51C. No transcriptome, mutation or epigenetic results are indicated in this manuscript, instead the authors state that HRD analysis was performed according to Popova et al 2012, but in this paper there is no mention of HRD as a measurable parameter and no way for calculating HRD is described. Hence no clear methodology is presented for GI and HRD calculations.
  • In the revision cover letter in testing BRCA1,2 homologous deletions the results show that 10/11 patients are HRD positive. Therefore, the aCGH based HRD approach used in this report might miss around 9% of BRCA1/2 depleted tumors. Although the number of patients is small this needs to be discussed.
  • Did the authors utilize the GISTIC2 to estimate focal SCNAs (deletions or amplifications involving a region smaller than 50% of a chromosome arm) as Davoli et al did?
  • In figure 3 although mentioned in the results section the frequencies of total chromosome SCNA are not depicted.
  • How the optimal cutoff of 88 was estimated for the GI index?
  • The authors must provide R scripts and full details of each application.

Minor comments

Lines 343-344:

“…All results are available in supplental figure 3…” change with “…All results are available in supplemental figure 3…”

Lines 377-378:

“…These data suggest that beyond genomic instability, genotoxique stress (by MMR deficiency or HR deficiecy) could induce hyper-expression of HLA-E…”

Change with

“…These data suggest that beyond genomic instability, genotoxic stress (by MMR deficiency or HR deficiency) could induce hyper-expression of HLA-E…”

Round 3

Reviewer 1 Report

The methodology used to assay genomic instability solely based on log-ratio thresholds from aCGH datasets is better clarified and properly referenced in the 2nd revised version of the manuscript, but still raises significant skepticism. For example, the authors stated in the 1st cover letter, that through their methodology, they assessed homozygous deletions of BRCA1-2 in about 16% of 67 HGSOC patients (11 out of 67). However, a shallow search in the cBioPortal for Cancer Genomics indicates that analysis of a much larger-scale cancer genomics data set of high-grade ovarian cancers from Pancancer Atlas, based on SNPs, reveals that deep deletions of these two genes do not exceed 2%. Furthermore, homozygous BRCA1-2 deletions imply that no BRCA1-2 genes are present and thus no protein is expected to be produced to check mutational status or perform functional analysis. The authors must state in the discussion that the way they have calculated GI and HRD scores solely through aCGH data, neglecting ploidy and tumor purity provides only a rough estimation on genomic instability.

Author Response

Dear Reviewer 1,

We thank you for your opinion. We have added a sentence in the discussion to insist on the methodological limits of the analysis of genomic instability by CGH. 

"The calculation of the HRD score was performed on CGH data not considering ploidy or tumor purity. This method gave an estimate of genomic instability."

King regards